# Clinical Course of Pseudophakic Cystoid Macular Edema Treated with Nepafenac

**DOI:** 10.3390/jcm9093034

**Published:** 2020-09-21

**Authors:** Alexander Aaronson, Asaf Achiron, Raimo Tuuminen

**Affiliations:** 1Helsinki Retina Research Group, University of Helsinki, FI-00290 Helsinki, Finland; amaaronson@gmail.com; 2Department of Ophthalmology, Helsinki University Hospital, FI-00290 Helsinki, Finland; 3Department of Ophthalmology, The Edith Wolfson Medical Center, 58100 Holon, Israel; achironasaf@gmail.com; 4Department of Ophthalmology, Sackler School of Medicine, Tel Aviv University, 69978 Ramat Aviv, Israel; 5Kymenlaakso Central Hospital, Unit of Ophthalmology, Kotkantie 41, FI-48210 Kotka, Finland

**Keywords:** cataract surgery, consensus criteria, intravitreal injection, non-steroidal anti-inflammatory drug, pseudophakic cystoid macular edema

## Abstract

Background: To evaluate the clinical course of pseudophakic cystoid macular edema (PCME) treated with topical non-steroidal anti-inflammatory drugs (NSAIDs). Methods: An analysis of the clinical course of PCME consisting of 536 eyes of 536 patients from five consecutive randomized clinical trials aimed at the optimization of anti-inflammatory medication in patients undergoing routine cataract surgery. PCME was classified as (i) grade 0a; no macular thickening, (ii) grade 0b; macular thickening (central subfield macular thickness (CSMT) increase of at least 10%) without signs of macular edema, (iii) grade I; subclinical PCME, (iv) grade II; acute PCME, (v) grade III; long-standing PCME. Eyes with PCME classification from grade I onwards were treated with nepafenac 1 mg/mL t.i.d. for two months. Results: CSMT increase of at least 10% at any postoperative timepoint with cystoid changes—a criterion for PCME—was found in 19 of 536 eyes (total incidence 3.5%). Of these 19 eyes, 13 eyes (total incidence 2.4%) had clinically significant PCME. PCME was considered clinically significant when both of the following visual acuity criteria were fulfilled. At any timepoint after the cataract surgery both the corrected distance visual acuity (CDVA) gain was less than 0.4 decimals from that of preoperative CDVA, and the absolute CDVA level remained below 0.8 decimals. Only one of the 19 eyes with criteria for PCME (total incidence 0.2%, incidence of PCME eyes 5.3%) showed no macular edema resolution within 2 months after topical nepafenac administration. **Conclusions:** PCME in most cases is self-limiting using topical nepafenac without any further need for intravitreal treatment.

## 1. Introduction

The reported incidence of pseudophakic cystoid macular edema (PCME) varies greatly depending on the methods used to identify it, and the diagnostic criteria used to classify its occurrence after cataract surgery. Advances in surgical equipment have minimized the need for intraoperative manipulation. Concurrently, identification of patients at risk for PCME and optimization of anti-inflammatory medication has reduced the incidence of clinically significant PCME to between 0.1 and 2.3% [1,2]. A large database analysis placed the incidence at 1.17% in patients with steroid monotherapy when no operative complication or recognized risk factors were present [3].

Diabetes [1,3,4,5,6,7,8], uveitis [1,9], epiretinal membrane [1,7,10], prior history of macular hole and contralateral PCME [7], retinal vein occlusion [3,7], intraoperative capsule rupture with or without vitreous loss, and intraoperative iris manipulation [3,11] are known to increase the incidence of PCME. Furthermore, prostaglandin analogs (PG) in the treatment of glaucoma are suspected to cause a higher incidence of PCME [12], but the evidence is debated as other studies have failed to recognize preoperative PG use as a risk for PCME [3,7]. Interestingly, in patients with diabetes, transient corneal edema was found to be predictive for PCME development [13].

Treatment recommendations for PCME also vary greatly. The use of intravitreal anti-VEGFs is controversial. Barone et al. identified a positive effect with intravitreal bevacizumab [14], while Splitzer et al. illustrated the effect to be nonexistent [15]. In studies of bevacizumab-resistant cases, the use of a dexamethasone intravitreal implant improved corrected distance visual acuity (CDVA) and reduced central foveal thickness (CFT) [16,17]. Triamcinolone acetonide (TA) was shown to mediate similar effects compared to the dexamethasone implant [18,19,20], while other studies have recognized the need for more frequent injections due to the more temporary effect than with the dexamethasone implant [21,22]. Repetition of TA injections accompanied increased intraocular pressure, whereas with the dexamethasone implant it achieved a plateau and gradually decreased thereafter [23]. The use of TA via subconjunctival or periocular administration is commonly practiced for prophylaxis and treatment of PCME [24], but a review by Han et al. recognized that this practice lacks strong scientific support [25]. However, the use of subconjunctival TA in diabetic patients at the end of cataract surgery reduced macular volume and thickness at 6 and 12 weeks, while bevacizumab had no effect [26].

Here, we evaluate the clinical course of PCME treated with the topical non-steroidal anti-inflammatory drug (NSAID) nepafenac.

## 2. Materials and Methods

This study is a retrospective analysis of five consecutive randomized clinical trials (RCTs) aiming at the optimization of anti-inflammatory medication in patients undergoing routine cataract surgery, conducted at the Kymenlaakso Central Hospital, Kotka, Finland (EU Clinical Trials Register Numbers: 2015-003296-30; 2015-005313-79; 2016-004514-10; 2016-004515-12; 2016-004784-40) [24,27,28,29,30]. The patients were admitted according to the national guidelines for the management of cataracts, and they were enrolled between January 2016 and December 2017. The corresponding author was the principal investigator and corresponding author on all five of these RCTs.

The RCTs were performed single- or double-masked when applicable. In the first study, we compared the efficacy of steroids, nonsteroidal anti-inflammatory drugs (NSAIDs), and their combination in 189 eyes of 180 patients undergoing routine cataract surgery [27]. In the second study, we compared the tolerability of two potent NSAIDs in 96 eyes of 95 patients also undergoing routine cataract surgery [28]. In the third study, we examined the role of preoperative anti-inflammatory treatment in the recovery from routine cataract surgery in 103 eyes of 103 diabetic patients treated with a combination of anti-inflammatory drugs [29]. In the fourth study, we compared the efficacy, safety, and tolerability of a perioperative subconjunctival triamcinolone acetonide injection with topical steroid drops in 109 eyes of 103 patients undergoing routine cataract surgery [24]. In the fifth study, we compared the efficacy of steroids, NSAIDs, and their combination in 60 eyes of 60 patients with pseudoexfoliation syndrome also undergoing routine cataract surgery [30]. To avoid bias from statistical dependence, in patients recruited for both eyes, only the first-operated eye was included in the post hoc analysis.

Aqueous flare, CDVA, central subfield macular thickness (CSMT), and intraocular pressure (IOP) were recorded. The studies were conducted according to the tenets of the Declaration of Helsinki and were approved by the Research Director and Chief Medical Officer of the Kymenlaakso Central Hospital, the Finnish Medicines Agency Fimea, and the Institutional Review Board of the Helsinki University Hospital. All patients signed an informed consent form before enrollment and could withdraw from the study at any time.

### 2.1. Patients

All studies were conducted as prospective single-center trials (hrrg.fi/en/clinicaltrials/cataract/). The patients were randomized by a research technician for different anti-inflammatory medication protocols according to the study plan. To avoid bias from statistical dependence, in patients recruited for both eyes, only the first-operated eye was included in the post hoc analysis. For detailed inclusion and exclusion criteria of the study subjects and the standardized surgical technique please find Supplemental Material and Methods.

### 2.2. PCME Classification

The operating physician examined the patients preoperatively. Postoperative measurements were performed masked from the treating physicians. CSMT (defined as mean thickness in the central 1000-μm diameter area) was recorded by spectral-domain optical coherence tomography (SD-OCT; Heidelberg Eye Explorer Version 1.9.10.0 and HRA/SPECTRALIS^®^ Viewing Module Version 6.0.9.0, Heidelberg Engineering GmbH, Heidelberg, Germany). Follow-up 30-frame SD-OCT scans were performed with AutoRescan^TM^ software.

The diagnosis of PCME was confirmed together with two physicians in all the studies. A 10% increase in CSMT was used as the cut-off value for macular thickening. The criteria for PCME were no pre-existing macular edema on preoperative OCT. PCME was classified as following: grade 0a; no macular thickening (CSMT < 10% from the baseline), grade 0b; macular thickening (CSMT ≥ 10% from the baseline, no signs of macular edema), grade I; subclinical PCME (CSMT ≥ 10% from the baseline, and cystoid changes near the fovea at any postoperative time point), grade II; acute or clinically significant PCME (grade I criteria, and expected deterioration of CDVA at any postoperative timepoint; PCME was considered clinically significant when both of the following VA criteria were fulfilled. At any timepoint after the cataract surgery both the CDVA gain was less than 0.4 decimals from that of preoperative CDVA, and the absolute CDVA level remained below 0.8 decimals.

Grade III; long-standing PCME (grade II criteria, and no resolution of macular edema within 2 months with topical NSAID treatment).

### 2.3. Statistical Analyses

Data are given as mean ± standard deviation, except for the absolute numbers and proportions for the nominal scale. IBM SPSS Statistics 25 (SPSS Inc., Somers, NY, USA) was used for statistical analysis. For two group comparisons, the Mann–Whitney U test was used. *P* ≤ 0.05 was considered statistically significant.

## 3. Results

### 3.1. PCME Incidence

Macular thickening, defined as a CSMT increase of at least 10% from the baseline, together with cystoid changes near the fovea—a criterion for PCME—was identified in 19 of the 536 patients, representing a total incidence of 3.5% (Table 1). Of the PCME cases, expected deterioration of CDVA at any postoperative timepoint (acute PCME) was found in 13 cases and long-standing PCME was found in one case, representing total incidence of 2.4% and 0.2%, respectively (Table 1). Representative OCT scans are shown in Figure 1.

Baseline patient and ophthalmic variables and surgical parameters did not differ between patients without PCME and any subset of patients with PCME (Table 2).

### 3.2. Clinical Outcomes according to the PCME Classification

At 28 days, in the eyes with PCME ≥ grade 1, the CSMT increase was higher (+98.1 ± 95.7 μm vs. +4.0 ± 22.1 μm, *P* < 0.001, Table 3) and that of CDVA was lower (0.74 ± 0.27 decimals vs. 0.87 ± 0.29 decimals, *P* = 0.004, Table 3) when compared to the eyes without PCME.

At 28 days, in the eyes with PCME ≥ grade 2, the CSMT increase was higher (+129.9 ± 99.3 μm vs. +4.0 ± 22.1 μm, *P* < 0.001, Table 3) and that of CDVA was lower (0.64 ± 0.18 decimals vs. 0.87 ± 0.29 decimals, *P* = 0.008, Table 3) when compared to the eyes without PCME. Furthermore, in the eyes with PCME ≥ grade 2, aqueous flare increase was higher (+10.1 ± 9.9 pu/ms vs. +4.9 ± 15.5 pu/ms, *P* = 0.019, Table 3) when compared to the eyes without PCME. No difference in intraocular pressure (IOP) was recognized between the groups (Table 3).

Macular thickness parameters are presented in Table 4. Foveal thickness and maximum thickness at the central subfield area corresponding to 1000 µm diameter around the fovea correlated with the PCME classification, whereas parafoveal and perifoveal thickness did not.

Correlation between operation time, cumulative phaco energy, and changes in macular thickness (CSMT) are presented in Table 5. Operation time and cumulative phaco energy did not correlate with macular thickness changes.

## 4. Discussion

The lack of well-designed randomized clinical trials for PCME has been recognized [31]. In particular, it would be important to differentiate and optimize a treatment protocol for long-standing refractory forms of PCME to set evidence-based guidelines.

We found that the majority of PCME cases treated with topical nepafenac were self-limiting without the need for intravitreal drug administration. Our observations form the basis and justification for interventional prospective trials between current clinical practice in some units, intravitreal injections, and topical NSAIDs.

The study designs of the trials in which PCME is analyzed typically comprise uncomplicated cataract surgeries. However, longer surgical time and higher applied cumulative phaco energy could hypothetically increase the risk for PCME development, but such a correlation was not identified.

Topical use of NSAIDs with or without topical glucocorticoids (GCs) was shown to reduce the risk of PCME development in a Cochrane meta-analysis [32] and a large multicenter study [33]. In minimizing PCME incidence, the superiority of NSAIDs over GCs was recognized [27,34]. NSAIDs have been compared head-to-head for anti-inflammatory efficacy, for instance diclofenac 0.1% versus nepafenac 0.1%, but no clinically meaningful difference has been recognized [20]. Although there is a growing tendency to prefer NSAID over GC, a recent letter highlighted how the comparison between NSAIDs and GCs is skewed towards comparing weaker GCs, often not used in clinical practice, with NSAIDs [35]. Moreover, the use of a periocular triamcinolone injection with diabetic patients reduced the incidence of PCME to zero [18], while bevacizumab had no effect [18].

In refractory PCME cases, the use of bevacizumab reduced macular thickness and improved BCVA in retrospective and prospective non-comparative studies [36]. In addition, in PCME topical triamcinolone-liposomes [37], interferon alpha 2b [38], interferon alpha 2a taken systemically [39], and the use of intravitreal infliximab [40] showed promising anatomical and functional responses to the treatment. To avoid overtreatment, based on the results presented here, we do not recommend the routine use of intravitreal or periocular injective treatments, nor systemic drug administration. Instead, the majority of acute PCME cases were self-limiting with topical nepafenac and applying the “wait and see” strategy. After 2 months of the treatment, PCME persisted only in one eye which responded well to the intravitreal injection of TA 4 mg. As such, it is favorable to formulate a strategy to find treatment-resistant PCME, otherwise we put our patients at a higher risk of complications with overtreatment, especially in the case of intravitreal treatment strategies. If a complication of surgery seems to be self-limiting in most of the cases, this is good news for the patient when the operation risks are evaluated prior to surgery. In any other scenario, all the necessary means would be applied to improve the patients’ CDVA and well-being.

In this study, clinically significant PCME correlated with the changes in the foveal thickness and within 1000µm diameter area. On the other hand, parafoveal and perifoveal sectors corresponding to 1–3 mm and 3–6 mm diameter area, respectively, did not correlate with PCME. At 28 days, aqueous flare increase from the baseline was significantly higher in the acute and long-lasting PCME subgroup (grade 2+) compared to those without PCME (grade 0). This finding is in accordance with the previous observation that flare values could be a feasible method to predict PCME [41]. The authors recognize that the low number of PCME reduces the reproducibility and statistical power of the results.

Taken together, differentiation and standardized criteria for refractory and long-lasting PCME are warranted both for research purposes for proper characterization of the complicated cases, as well as for clinical practice to better allocate treatment resources.

## Figures and Tables

**Figure 1 jcm-09-03034-f001:**
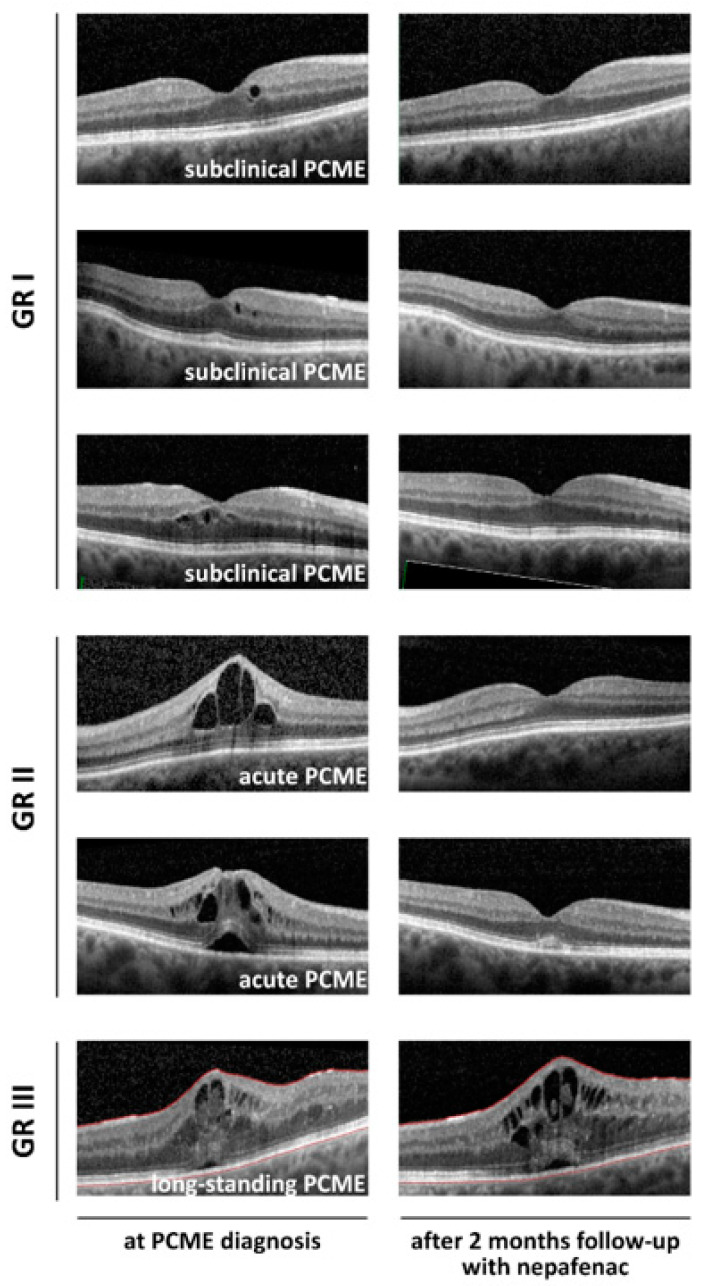
Representative OCT scans.

**Table 1 jcm-09-03034-t001:** Distribution of pseudophakic cystoid macular edema (PCME) within 3 months from cataract surgery.

	N = 536 Eyes (of 536 Patients)
PCME criteria: No pre-existing ME on preoperative OCT	
**PCME classification**	
**GR 0a** = no retinal thickening (CSMT < 10%, no signs of ME)	*N* = 498 (93%)
**GR 0b** = retinal thickening without ME (CSMT ≥ 10%, no signs of ME)	*N* = 19 (3.5%)
**GR I** = subc. PCME (CSMT ≥ 10% at any postoperative timepoint + ME)	≥GR I *N* = 19 (3.5%)
**GR II** = acute PCME (GR I + expected CDVA deterioration)	≥GR II *N* = 13 (2.4%)
**GR III** = l-s PCME (GR II + no ME resolution within 2 mo/with topical meds)	GR III *N* = 1 (0.2%)

Data are given as absolute numbers and proportions. Expected deterioration of CDVA is defined as both CDVA gain below 0.4 decimals from the baseline, and postoperative CDVA below 0.8 decimals at any postoperative timepoint. CDVA; corrected distance visual acuity, l-s; long-standing, ME; macular edema, CSMT; mean central subfield macular thickness, OCT; optical coherence tomography, subc; subclinical.

**Table 2 jcm-09-03034-t002:** Baseline (preoperative) parameters according to the PCME classification.

	Gr 0	≥GR I	≥Gr II	=GR III
	No ME	Subclinical PCME	Acute PCME	Long-Standing PCME
Age (years)	76.1 ± 6.7	76.4 ± 7.1	77.2 ± 7.6	77
Male:Female (%)	35:65	36:64	56:44	1:0
Operation time (min)	19.1 ± 9.1	20.5 ± 8.6	22.2 ± 8.9	22
Phacoenergy (CDE)	19.9 ± 10.6	23.0 ± 11.5	22.8 ± 11.8	31.88
Aqueous flare (pu/ms)	9.7 ± 8.0	11.4 ± 8.1	9.6 ± 5.8	6.3
CSMT (μm)	273.5 ± 32.4	274.4 ± 34.9	286.9 ± 42.2	312
CDVA (decimals)	0.36 ± 0.17	0.35 ± 0.14	0.35 ± 0.16	CF
IOP (mmHg)	15.8 ± 4.0	14.5 ± 3.9	14.1 ± 4.5	10

Data are given as mean ± standard deviation (±SD). CDVA; corrected distance visual acuity, CF; counting fingers, CSMT; mean central subfield macular thickness, IOP; intraocular pressure, pu; photon units.

**Table 3 jcm-09-03034-t003:** Clinical outcomes according to the PCME classification.

	Gr 0	≥GR I	≥Gr II	=GR III
	No ME	Subclinical PCME	Acute PCME	Long-Standing PCME
**Aqueous flare** (pu/ms)				
change at 28 days	+4.9 ± 15.5	+8.4 ± 9.8	+10.1 ± 9.9 *	+29.2
**CSMT** (μm)				
change at 28 days	+4.0 ± 22.1	+98.1 ± 95.7 ^†^	+129.9 ± 99.3 ^†^	+155
**CDVA** (decimals)				
at 28 days	0.87 ± 0.29	0.74 ± 0.27 *	0.64 ± 0.18 *	0.50
**IOP** (mmHg)				
at 28 days	10.6 ± 2.8	11.0 ± 3.2	10.7 ± 3.5	8

Data are given as mean (±SD). CDVA; corrected distance visual acuity, CSMT; mean central subfield macular thickness, IOP; intraocular pressure, pu; photon units. * *P* < 0.05; ^†^
*P* < 0.001.

**Table 4 jcm-09-03034-t004:** Macular thickness parameters.

	Gr 0	≥GR I	≥Gr II	=GR III
	No ME	Subclinical PCME	Acute PCME	Long-Standing PCME
**Foveal thickness**				
at 28 days (μm)	247.5 ± 60.6	286.0 ± 138.6	327.8 ± 155.7	439
change at 28 days (μm)	6.36 ± 49.6	45.2 ± 109.5	83.0 ± 130.8 *	228
change at 28 days (%)	3.76 ± 23.3	16.8 ± 44.1	33.3 ± 55.4 *	108.1
**CRT max**				
at 28 days (μm)	332.6 ± 68.7	364.5 ± 105.7	406.8 ± 136.4	612
change at 28 days (μm)	9.6 ± 60.7	41.2 ± 76.5	78.6 ± 118.8 *	314
change at 28 days (%)	3.16 ± 18.8	11.5 ± 20.8*	23.4 ± 37.0 *	105.4
**Parafoveal thickness**				
at 28 days (μm)	333.1 ± 29.0	333.6 ± 54.7	347.1 ± 54.5	426
change at 28 days (μm)	7.54 ± 19.6	4.29 ± 36.2	17.2 ± 27.1	37.5
change at 28 days (%)	2.32 ± 6.02	0.96 ± 10.6	4.82 ± 7.41	9.66
**Perifoveal thickness**				
at 28 days (μm)	291.5 ± 20.2	293.0 ± 25.4	291.2 ± 27.7	339.7
change at 28 days (μm)	5.50 ± 15.7	4.71 ± 9.42	8.37 ± 4.58	10.4
change at 28 days (%)	2.01 ± 4.96	1.69 ± 2.98	2.94 ± 1.53	3.16

Data are given as mean ( ±SD). CRT max; maximum thickness at central subfield area corresponding to 1000 μm diameter around the fovea. * *P* < 0.05.

**Table 5 jcm-09-03034-t005:** Surgical variables according to the PCME classification.

	Gr 0	≥GR I	≥Gr II	=GR III
	No ME	Subclinical PCME	Acute PCME	Long-Standing PCME
Operation time (min)	18.9 ± 8.8	22.1 ± 8.7	21.9 ± 9.2	22
Phacoenergy (C.D.E.)	20.2 ± 11.0	23.2 ± 11.6	21.2 ± 10.9	31.88

Data are given as mean (±SD). C.D.E.; cumulative dissipated energy.

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
