# Peer review of "Clinical Course of Pseudophakic Cystoid Macular Edema Treated with Nepafenac"

_jcm, 2020, doi:10.3390/jcm9093034_

Round 1

Reviewer 1 Report

Re-review jcm-911000, Clinical course of pseudophakic cystoid macular edema treated with nepafenac

Page 1, Line 24-5: ‘both CDVA gain below 0.4 decimals from baseline and post-operative CDVA below 0.8 decimals’. The meaning of this sentence is quite obscure to me. Do the authors mean that the gain of vision was related to decimal 0.4 or 0.8 absolutely, or a change in several lines of vision on the visual acuity chart?

Page 1, Line 37: ‘clinically significant PCME in between’ should read ‘clinically significant PCME to between’’.

Page 2, Line 65: ‘This study is a retrospective analysis of five consecutive randomized clinical trials’. The authors should state clearly that at least one of them was a co-author on all five of these RCTs.

Page 3, Line 128-9: ‘Representative OCT scans are shown in Figure 1.’ The Authors need to state precisely what the nature and severity (degree) of the PCME in each of these photographs of Figure 1. Presumably the right-hand column is the normal, contralateral eye, though it might be the healed eye of the left-hand figure after being treated with nepafenac. The choroidal circulation gives a clue that this is indeed the same eye after treatment, but that has to be stated.

In Table 3, flare is positively correlated with PCME. Therefore, the authors should clearly state that Table 2 ‘Baseline’ means PREOPERATIVE flare status.

As previously mentioned, I would have expected all patients whose PCME resolved, unless there were other anatomical abnormalities, to have achieved decimal VA of 1.6 in over 90% of cases. If the authors do not have knowledge of the final visual acuities, which is what treatment of PCME is all about, they cannot make a recommendation that it does any good, other than that it makes the OCT look a little more agreeable.

Page 5, Line 151-152: ‘Operation time and cumulative phaco energy did not correlate with 151 macular thickness changes’. I have always felt that this is the case.

I consider that the aim should be to do a phaco procedure as technically perfectly as possible.  In my own multiple published reports evaluating over 5,000 cases, studied prospectively and consecutively with no exclusions, CDVA very close to the decimal 1.6 was achieved in over 93% of cases. In those that did not, an explanation was found in the remaining 7% of cases.

Some colleagues are revered if they can do a case in 5-6 minutes. That is fine by me, as long as the CDVA at one month is around 1.6. If it is worse, an explanation must be provided. In the worst case scenario, such surgeons may legitimately be asked to operate more slowly and more carefully, doing everything they can to achieve CDVA of 1.6 at one month.

Page 6, Line 162-3: ‘Our observations lay the basis and justification for interventional prospective trial between current clinical practice in some units, intravitreal injections and topical NSAIDs.’ I agree with this wholeheartedly. In fact, with the Authors’ track record of these five studies, they could almost certainly persuade the Ethics Committee of their hospital to allow them to carry out such a prospective study. I urge them to do so.

Page 2, from the comments to Reviewer 3: ‘Here, our PCME cases were defined as “expected deterioration of CDVA at any post-operative timepoint; both CDVA gain below 0.4 decimals from the baseline and post-operative CDVA below 0.8 decimals”.

Typically, PCME deteriorates the CDVA few decimals, in the major case, grade III example, CDVA at 1-month was 0.5’. Here the authors have stated that the presumed difference of Visual acuity could be a little better than 20/50, but surely this depends on what it was before the surgery. I do not understand this description. The authors appear to be using a differential change of VA between two time points, rather than an ultimate CDVA or gaining of several lines of acuity.

Author Response

We thank the Reviewer for the thorough consideration of our article. We hope to satisfy all the necessary concerns.

Page 1, Line 24-5: ‘both CDVA gain below 0.4 decimals from baseline and post-operative CDVA below 0.8 decimals’. The meaning of this sentence is quite obscure to me. Do the authors mean that the gain of vision was related to decimal 0.4 or 0.8 absolutely, or a change in several lines of vision on the visual acuity chart?

The authors defined CDVA criteria as follows: both i) the CDVA improved less than 0.4 decimals from that of preoperative CDVA, and ii) the CDVA remained at the level below 0.8 decimals after cataract surgery. We have now revised this section in the Abstract (page 1 lines 23-6) and in the Methods (page 3 lines 116-8) of the manuscript.

Page 1, Line 37: ‘clinically significant PCME in between’ should read ‘clinically significant PCME to between’’.

We agree with this consideration and the text has been changed: “clinically significant PCME to between”. (page 1 line 39)

Page 2, Line 65: ‘This study is a retrospective analysis of five consecutive randomized clinical trials’. The authors should state clearly that at least one of them was a co-author on all five of these RCTs.

This is an important statement. We have added the next statement: “The corresponding author has been the principal investigator and corresponding author on all five of these RCTs.” (page 2 lines 71-2)

Page 3, Line 128-9: ‘Representative OCT scans are shown in Figure 1.’ The Authors need to state precisely what the nature and severity (degree) of the PCME in each of these photographs of Figure 1. Presumably the right-hand column is the normal, contralateral eye, though it might be the healed eye of the left-hand figure after being treated with nepafenac. The choroidal circulation gives a clue that this is indeed the same eye after treatment, but that has to be stated.

Thank you for the careful revision. Indeed, right-hand figures represent healed eye of the left-hand figure after being treated with nepafenac. We have now clarified our Figure 1.

In Table 3, flare is positively correlated with PCME. Therefore, the authors should clearly state that Table 2 ‘Baseline’ means PREOPERATIVE flare status.

Again, this is an important statement. We have added the following correction in the title of Table 2:

Table 2. Baseline (preoperative) parameters according to the PCME classification.

As previously mentioned, I would have expected all patients whose PCME resolved, unless there were other anatomical abnormalities, to have achieved decimal VA of 1.6 in over 90% of cases. If the authors do not have knowledge of the final visual acuities, which is what treatment of PCME is all about, they cannot make a recommendation that it does any good, other than that it makes the OCT look a little more agreeable.

This is a very good viewpoint we haven’t considered and might be true. So far, in all our RCTs we have evaluated CDVA up to 1.0 (which is a common national practice), but not any further. Indeed, VA testing up to 2.0 would give us more information on the postoperative visual acuity status. In the future trials, we will improve this detail to achieve even better understanding on the visual acuity. 

Page 5, Line 151-152: ‘Operation time and cumulative phaco energy did not correlate with macular thickness changes’. I have always felt that this is the case.

We agree that it is important to provide evidence regarding this matter. Thank you for these comments, Table 5 now improves the overall impression of the manuscript.

I consider that the aim should be to do a phaco procedure as technically perfectly as possible.  In my own multiple published reports evaluating over 5,000 cases, studied prospectively and consecutively with no exclusions, CDVA very close to the decimal 1.6 was achieved in over 93% of cases. In those that did not, an explanation was found in the remaining 7% of cases.

Some colleagues are revered if they can do a case in 5-6 minutes. That is fine by me, as long as the CDVA at one month is around 1.6. If it is worse, an explanation must be provided. In the worst case scenario, such surgeons may legitimately be asked to operate more slowly and more carefully, doing everything they can to achieve CDVA of 1.6 at one month.

Thank you. Your viewpoints are exactly on the spot with our message on the improvement of the quality of the surgery and its perioperative and postoperative management. As we can see, overall operation time averaged around 20 minutes in our surgeries, which is relatively long time.

Some of the surgeons operate on 6-8 minutes most of the eyes, but they seldom mention their complication rates, incidence of PCO formation, postoperative refractive results in premium IOLs. Even long-term IOL luxation rates in PXF eyes reflect the surgeon’s technique.

Our philosophy is the opposite. In another study of 14520 eyes [In Press] we have reviewed our surgical complications, education of the residents and technological advancements over the past 10 years.

I think we share the same ideology, that cataract surgery is not only the IOL implantation or even the surgical manoeuvre, but requires e.g. preoperative planning on optimal IOL selection, diligent operative care and optimal post-operative management to achieve best vision-related quality-of-life outcomes.

Page 6, Line 162-3: ‘Our observations lay the basis and justification for interventional prospective trial between current clinical practice in some units, intravitreal injections and topical NSAIDs.’ I agree with this wholeheartedly. In fact, with the Authors’ track record of these five studies, they could almost certainly persuade the Ethics Committee of their hospital to allow them to carry out such a prospective study. I urge them to do so.

Thank You for this comment. Our research group pursue such a prospective study given the experience on the subject. Acknowledging the relatively low incidence of the PCME and the necessary power (sample size) for randomization we aim to carry out this research plan in a large University Hospital site.

Page 2, from the comments to Reviewer 3: ‘Here, our PCME cases were defined as “expected deterioration of CDVA at any post-operative timepoint; both CDVA gain below 0.4 decimals from the baseline and post-operative CDVA below 0.8 decimals”.

Typically, PCME deteriorates the CDVA few decimals, in the major case, grade III example, CDVA at 1-month was 0.5’. Here the authors have stated that the presumed difference of Visual acuity could be a little better than 20/50, but surely this depends on what it was before the surgery. I do not understand this description. The authors appear to be using a differential change of VA between two time points, rather than an ultimate CDVA or gaining of several lines of acuity.

We agree that the VA criteria is written complicated in its current form,

instead of

“..were found with expected deterioration of corrected distance visual acuity (CDVA) at any post-operative timepoint; both CDVA gain below 0.4 decimals from baseline and post-operative CDVA below 0.8 decimals”.

we have now revised our sentence

 “PCME was considered clinically significant when both of the following VA criteria fulfilled. At any timepoint after the cataract surgery both the CDVA gain was less than 0.4 decimals from that of preoperative CDVA, and the absolute CDVA remained at the level below 0.8 decimals”.

Reviewer 2 Report

Addressed comments adequately.

Author Response

We thank the Reviewers #2 for excellent comments and remarks that have helped us to improve the manuscript

Reviewer 3 Report

In this manuscript, authors reported natural course of patients with catract surgery following nepafenac. Readers will notice chronic Grade III incidence rate is 0.2%. The rate is important information, but readers' most interest might be how to care the CME after catract surgery.

The manuscript is well written and I don't feel any improvement points.

Author Response

We thank the Reviewers #3 for excellent comments and remarks that have helped us to improve the manuscript.

This manuscript is a resubmission of an earlier submission. The following is a list of the peer review reports and author responses from that submission.

Round 1

Reviewer 1 Report

This is a well constructed report of an important clinical subject.

Despite no operative complications during cataract surgery a considerable number of individuals will develop cystoid macular edema. Many more cases are found on OCT testing of the macula than are noticed based on visual acuity testing without OCT. As a cataract surgeon I am well aware of this issue. Topical corticosteroids and non-steroidal anti-inflammatory agents have been used in drop form or injected into the anterior chamber at the time of surgery to reduce the incidence of this complication. Both types of pharmacologic agents have long been used. NSAIDS are preferred since they do not raise the intra-ocular pressure, while steroids have a higher incidence of raising the pressure. Many studies have been published about this issue, but few have been randomized clinical trials. The fact that this manuscript describes RCT data is important. In this study only one eye in the series of 536 was found to have persistent edema after two months of use of the non-steroidal anti-inflammatory agent nepafenac, quite good results. In summary, this well designed and well written study merits publication.

Reviewer 2 Report

The authors state the purpose of this study is "to evaluate the clinical course of pseudophakic cystoid macular edema (PCME) treated with topical non-steroidal anti-inflammatory drugs (NSAIDs)." There have been numerous studies that have documented the natural clinical course for PCME. The "novelty" of this study was to study the clinical course of PCME in pts treated with topical NSAID. While there are publications on clinical trials that have already commented on this, the authors undergo the extra step of examining multiple clinical studies. However, there are two major weaknesses with methodology, resulting in an unfavorable review.

1) The authors perform a retrospective study with 5 consecutive randomized clinical trials. However, the incidence of PCME is low; and therefore, the sample size is small. With a sample size this small, there's a concern for low statistical power, inflated false discovery rate, inflated effect size estimation, and low reproducibility.

2) PCME has been shown to be affected by surgical time, phaco power, and intraoperative complications. The authors do not control for this factor in their analysis, which diminishes the robustness and reliability of their conclusions. For example, in table 2, the patient with grade 3 PCME had higher phacoenergy used during surgery, which may have impacted his/her clinical course.

Given the simplicity of the authors' aim, to evaluate the natural course of PCME in pts treated with topical NSAIDS, it's critical that the data and conclusions are well-supported and robust. Unfortunately, the methodology as it stands does not seem to substantiate this requirement.

Reviewer 3 Report

Materials and Methods

The exclusion criteria indicate that although the studies were theoretically prospective, they were done in absolutely pristine eyes.

We looked at the five studies, and it was completely unclear from the in-hospital data what the results were. Presumedly the authors were looking only at the PCME rate.

This study needs to be performed prospectively with a surgeon who can ask a masked colleague/s to look at CDVA and CMST, at each postoperative time point, in a study that is large enough for statistically significant data. I do not think that looking at five in-hospital studies without full availability of data including CDVA is appropriate for making decisions about management of PCME.

Results

The first two paragraphs on page 7:

At 28 days, in eyes with PCME ≥ grade 1, the CSMT increase was higher…

At 28 days, in eyes with PCME ≥ grade 2, the CSMT increase was higher (+129.9 ± 99.3 m vs. +4.0 ± 22.1 m…

add nothing at all. The readership would already be cognisant of these findings.

Discussion

Page 8: ‘The lack of well-designed randomized clinical trials for PCME has been recognized [34]’: This Reviewer absolutely agrees. Therefore, it would have been much more insightful of the authors to have run such a trial themselves.

Page 9, paragraph 1, line 7: ‘Instead, the majority of acute PCME cases were self-limiting with topical nepafenac and applying the “wait and see” -strategy’:

As a surgeon myself, I do not like to be explaining to the patient why his or her vision of decimal 0.3 or worse is satisfactory, and that while it may improve, there is no guarantee that it will, especially untreated.

Considering the retinal histopathology involved, eminently demonstrated by the authors OCT figures, this surgeon can not really imagine how such a laissez-faire approach is appropriate.

In fact, although it is difficult to get a CDVA at one day or one month because of day one pupillary mydriasis and occasional folds in the cornea, visual acuity at every postoperative timepoint, namely, one day, one week and one month, should always be around 1.25 to 1.60 (20/15 to 20/12.5). If not, it should behove the surgeon to do the best possible for the patient, and to achieve that level of CDVA.

Page 9, paragraph 2, line 3: ‘At 28 days, aqueous flare increase from the baseline was significantly higher in acute and long-lasting PCME subgroup (grade 2+) compared to those without PCME (grade 0)’:

This Reviewer can not fathom why a patient with a prominent aqueous flare which has persistent since surgery is being tolerated.

In this Reviewer’s view, a prospective study done with a reasonably-sized cohort should be done examining PCME and managing it accordingly.